# Shape Transformation Approaches for Fluid Dynamic Optimization

**Peter Marvin Müller *** , **Georgios Bletsos** and **Thomas Rung**

Institute for Fluid Dynamics and Ship Theory, Hamburg University of Technology, Am Schwarzenberg-Campus 4, 21073 Hamburg, Germany; george.bletsos@tuhh.de (G.B.); thomas.rung@tuhh.de (T.R.)
* Correspondence: peter.marvin.mueller@tuhh.de

**Abstract:** The contribution is devoted to combined shape- and mesh-update strategies for parameter-free (CAD-free) shape optimization methods. Three different strategies to translate the shape sensitivities computed by adjoint shape optimization procedures into simultaneous updates of both the shape and the discretized domain are employed in combination with a mesh-morphing strategy. Considered methods involve a linear Steklov–Poincaré (Hilbert space) approach, a recently suggested highly non-linear *p*-Laplace (Banach space) method, and a hybrid variant which updates the shape in Hilbert space. The methods are scrutinized for optimizing the power loss of a two-dimensional bent duct flow using an unstructured, locally refined grid that initially displays favorable grid properties. Optimization results are compared with respect to the optimization convergence, the computational effort, and the preservation of the mesh quality during the optimization sequence. Results indicate that all methods reach, approximately, the same converged optimal solution , which reduces the objective function by about 18% for this classical benchmark example. However, as regards the preservation of the mesh quality, more advanced Banach space methods are advantageous in comparison to Hilbert space methods even when the shape update is performed in Hilbert space to save costs. In specific, while the computational cost of the Banach space method and the hybrid method is about 3.5 and 2.5 times the cost of the pure Hilbert space method, respectively, the grid quality metrics are 2 times and 1.7 times improved for the Banach space and hybrid method, respectively.

**Keywords:** adjoint-based optimization; CAD-free shape optimization; mesh update methods; gradient-descent shape optimization; CFD





## 1. Introduction

The ever-growing advances in computational sciences have made simulation-based design an indispensable tool for many engineering industries dealing with applications of either fluid [1,2] or structural mechanics [3]. A crucial aspect of the design is the shape of the investigated device.

To this end, a variety of shape optimization methods have been developed to enable the efficient identification of optimal, or rather optimized, shapes that minimize (or maximize) the response, namely the *objective functional*, of the shape to a set of prescribed conditions. Such methods range from stochastic [4] (or global) to deterministic [5] (or local) procedures, with the latter usually requiring the gradient of the objective functional with respect to the shape. This paper is concerned with continuous adjoint- and gradient-based shape optimization methods. Adjoint-based techniques have successfully been applied in various areas of industrial applications [6–9] in order to compute shape sensitivities (shape derivatives) required by a gradient-based optimization method. The appeal of adjoint methods is that the computation of the derivatives is independent of the number of design or control variables of the problem, as this method does not require the derivative of the state variable with respect to the control.

Generally, such methods can be categorized into parameterized, where the shapes follow an a priori explicit parameterization—e.g., specified by a set of control points [10]—or parameter-free methods, where the control refers directly to the investigated shapes [11–13]. As regards the former, a common technique is to directly apply the computed sensitivity expressions within a shape optimization loop. On the other hand, when the control corresponds to the (non-parameterized) shape itself, an additional step is required to identify admissible descent directions and sequentially applicable shape updates. As an additional step, researchers have successfully applied discrete filtering of the shape sensitivity field [12–14] or more involved approaches such as the Steklov–Poincaré [15], Laplace–Beltrami [8] or the recently suggested *p*-harmonic descent method [16,17]. A general overview of such methods for engineering applications can be found in [14,18]. These methods aim to obtain a deformation vector field with a certain regularity on the boundary, which leads to feasible shape updates. The issue of regularity is not only important in the context of shape optimization but also in other areas, such as optimal flow control; see, e.g., [19,20], where the issue of regularity is discussed for inhomogeneous boundary data.

Regardless of the applied approach, a critical aspect of the optimization process is the preservation of the mesh quality during the optimization. To reduce the computational cost, a simple restart from the previous design is highly appreciated. In particular, one ideally wants to conserve the grid topology and update the shape and the grid using mesh morphing techniques, i.e., without the need for re-meshing. To this end, the paper discusses recent update techniques. Attention is restricted to strategies that simultaneously compute the shape and mesh updates, i.e., the Steklov–Poincaré (Hilbert space) method [15] and the *p*-Laplace (Banach space) method [17]. While the former is deemed more efficient, the latter is seen to preserve the quality of the mesh much better but is also harder to solve [17,18].

In view of a viable compromise between mesh quality preservation and required computational effort, we investigate a novel hybrid approach that combines elements of both strategies. In contrast to the previously established methods, the hybrid method utilizes the Steklov–Poincaré approach to efficiently compute the displacement vector field on the boundary of the shape and then employs a *p*-harmonic domain extension to compute the displacement on the nodes of the internal mesh. The motivation of this approach is to achieve shape updates that are computationally viable for engineering applications while at the same time preserving the quality of the employed mesh by extracting the positive characteristics of each ingredient approach. The concept shares ideas with the extension method suggested in [21].

Within this paper, we particularly focus on mesh quality aspects and computational efficiency. The application presented, refers to power-loss optimization for a 2D S-bent duct flow. The unstructured grid discretization involves locally refined quadrilateral control volumes and the solution of the primal and adjoint Navier–Stokes equations is obtained from a classical second-order accurate finite-volume method. Optimized shapes are assessed in terms of the convergence of the objective functional, the final shape, as well as the mesh quality by means of the orthogonality of the grid and the aspect ratio of the cells for the final configuration. Preserving the mesh quality is a necessary prerequisite for trustworthy optimization results and is thereby of importance. Moreover, related issues might even cause the divergence of the optimization in mesh morphing approaches.

The two-dimensional application considered in this paper refers to an optimization problem constrained by the stationary Navier–Stokes equations. However, the methods presented can be readily applied to most shape optimization problems (two or three-dimensional) constrained by a set of partial differential equations (PDEs), as long as a shape sensitivity can be computationally approximated. Additionally, the investigated methods are not restricted to stationary problems and may be extended to time-dependent problems as well. In the case that the domain and thus the control parameter does not change over time the directional derivative of the shape functional represents a time averaged sensitivity.

If the shape, however, is varying over time, the domain of definition of the deformation vector field is no longer the spatial domain but the space–time domain.

The remainder of the paper is organized as follows: Section 2 briefly presents the optimization method. Emphasis is given to the domain transformation problem and related novel aspects applied herein. The section closes with algorithms that outline the complete process. Section 3 is concerned with an exemplary application that renders the characteristics of the proposed methodology. The paper closes with conclusions and outlines future directions in Section 4. Within this publication, Einstein's summation convention is used for repeated lower-case Latin subscripts. Furthermore we denote the iterate for the primal and adjoint state variable as well as the descent direction with a superscript index $k$, e.g., $u^k$ for the descent direction of the $k$-th shape $\Omega_k$.

## 2. Mathematical and Computational Model

### 2.1. Optimization Method

In general, let $\Omega \subset \mathbb{R}^d$, $d = 2$ or $d = 3$, be a domain with Lipschitz boundary $\Gamma$, and $y$ be a physical state defined on $\Omega$. In the context of the application studied in this paper, we consider $\Gamma$ to be the union of an inlet ($\Gamma_I$), outlet ($\Gamma_O$), and wall ($\Gamma_W$) boundary, that is $\Gamma = \Gamma_I \cup \Gamma_O \cup \Gamma_W$. We consider shape optimization problems of the general form

$$\min_{\Omega, y} J(\Omega, y) \quad \text{subject to} \quad e(\Omega, y) = 0, \tag{1}$$

where $e(\Omega, y)$ denotes the PDE constraints on the state $y$, which in our case correspond to the Navier–Stokes equations and $J(\Omega, y)$ is a shape function. We assume that the state is unique on $\Omega$, and thus the control to state mapping $\Omega \mapsto y(\Omega)$ exists [22]. Therewith we obtain the reduced objective function $J(\Omega, y(\Omega)) =: j(\Omega)$ and in order to compute shape sensitivities for $j(\Omega)$ the domain has to be made variable. We follow the standard ansatz with a perturbation of the identity $\text{id} + tu$ where the descent direction is given by the vector field $u : \Omega \to \mathbb{R}^d$ with $u \in W^{1,\infty}(\Omega)$, cf. [23] (Section 2.8) and [24] (Chapter 2, Section 2.6). Then the transformed domain reads

$$(\text{id} + tu)(\Omega) := \left\{ x + tu \in \mathbb{R}^d : x \in \Omega \right\}. \tag{2}$$

With a suitable displacement field $u$ and a sufficiently small step size $t > 0$, the perturbation of the identity is invertible with bounded inverse [23]. The shape derivative of the reduced cost function $j(\Omega)$ is denoted by $j'(\Omega)u$ and fulfills the approximation condition [25]

$$j((\text{id} + tu)(\Omega)) = j(\Omega) + tj'(\Omega)u + o(t) \quad \text{for} \quad t \to 0. \tag{3}$$

In the following, we describe methods of how to obtain a descent direction $u$ such that

$$j'(\Omega)u < 0 \tag{4}$$

holds.

### 2.2. Descent Approaches to Simultaneously Update the Mesh and the Shape

First we consider a Steklov-Poincaré-type method which has been introduced in [26]. Therein the descent direction is obtained by solving a linear elasticity-like problem from structural mechanics where the shape sensitivity enters as right hand side as forcing term. The method is similar to the *Hilbertian extension and regularization* in [27] and [28] (Section 5.2). In a Hilbert space setting the gradient of $j(\Omega)$ gives a descent direction. Consider the Hilbert space $H$ with the inner product $a(\cdot, \cdot) : H \times H \to \mathbb{R}$. Then we obtain a descent direction $u \in H$ defined as the solution of the variational form

$$a(u, w) = -j'(\Omega)w, \quad \forall w \in H. \tag{5}$$

Of course, the direction depends on the choice of $a(\cdot, \cdot)$. The Hilbert space approach leads to regular vector fields which are defined on the whole computational domain. However, the obtained transformations might not give 'good' deformations of the internal mesh; that is, the resulting displacement fields do not sufficiently preserve mesh quality, regarding e.g., changes to the level of orthogonality or the aspect ratio. This is only a technical issue but it has a significant influence on the implementation and algorithmic realization of the approach. We therefore suggest to compute a $p$-harmonic domain extension of the resulting shape deformations by solving a non-linear elliptic equation containing the $p$-Laplace operator. This second approach builds on the idea proposed in [21] where the extension of the boundary deformation is extended via a non-linear convection diffusion problem into the domain. Here, we compute a descent direction and the domain extension in a two step process. First, the shape gradient is given by the solution of (5), and second, the movement of the discrete nodes within the domain is given by the solution of the Dirichlet problem

$$-\frac{\partial}{\partial x_i}\left(\left(\left(\frac{\partial \bar{u}_i}{\partial x_j}\right)^2\right)^{\frac{p-2}{2}}\frac{\partial \bar{u}_j}{\partial x_i}\right) = 0 \quad \text{in } \Omega,$$

$$\bar{u} = u \quad \text{on } \Gamma. \tag{6}$$

for $p > 2$ in a weak sense. Because the solution of (5) enters as Dirichlet data, the shape itself still is deformed by $u \in H$ and only the internal nodes of the mesh are affected. For the inner product in (5) we consider

$$a(u, w) = \int_\Omega \eta \frac{\partial u_i}{\partial x_j}\frac{\partial w_i}{\partial x_j}\, dx, \tag{7}$$

with the diffusivity [9]

$$\eta(x) = \frac{1}{\eta_{max}^{-1} + \min\limits_{\bar{x} \in \Gamma}\|x - \bar{x}\|_2}. \tag{8}$$

For now, we neglect the fact that in general $H \not\subset W^{1,\infty}(\Omega, \mathbb{R}^d)$ when considering (5) with the inner product (7). However, the obtained solutions give regular deformations and are applicable for practical use. In [15], this method is associated with applying the Dirichelt-to-Neumann map or Steklov–Poincaré operator for the case in which $j'(\Omega)u$ has a boundary formulation of the form

$$j'(\Omega)u = \int_{\Gamma_D} \sigma u_i n_i\, ds \tag{9}$$

where $\sigma : \Gamma \to \mathbb{R}$ depends on the state and the adjoint state and thus it is specific to the problem. Note that the formulation in (5) is rather general and does not necessarily require the boundary formulation of the shape derivative.

Third, we consider the $p$-Laplace relaxation of the steepest descent direction in $W^{1,\infty}$-topology [16,17]. The direction of steepest descent in $W^{1,\infty}(\Omega, \mathbb{R}^d)$ is defined by

$$u^* = \arg\min\limits_{u \in W^{1,\infty}(\Omega, \mathbb{R}^d),\, \|\nabla u\| \leq 1} j'(\Omega)u \tag{10}$$

where $\|\cdot\|$ is the operator (spectral) norm. Following [29] (Proposition 5.1 and 5.3) the minimizer of

$$I(u) := \frac{1}{p}\int_\Omega (\nabla u : \nabla u)^{p/2}\, dx + j'(\Omega)u \tag{11}$$

tends to the solution of (10) for $p \to \infty$ and thus the desired direction of steepest descent. The approximation is obtained by solving the boundary value problem

$$-\frac{\partial}{\partial x_i}\left(\left(\left(\frac{\partial u_k}{\partial x_l}\right)^2\right)^{\frac{p-2}{2}}\frac{\partial u_j}{\partial x_i}\right) = 0 \qquad \text{in } \Omega,$$

$$u = 0, \qquad \text{on } \Gamma \setminus \Gamma_D,$$

$$\eta\,\frac{\partial u_j}{\partial n} = -\sigma n_j, \qquad \text{on } \Gamma_D.$$

$$\tag{12}$$

In practice, this means that we are interested in solutions of (11) for $p$ as large as possible. This involves at least two difficulties for the practical application. On the one hand, the numerical computation of solutions of (11) requires higher computational effort the higher the value of $p$ is [30,31]. On the other hand, the order of integrability $p$ depends on the spatial dimension; that is, $p > d$ has to be fulfilled, which directly follows from the Sobolev imbedding theorem [32] (Theorem 4.12). Additionally, when considering a second order method for solving (11) the second derivative of $I(u)$ does not exist where $\nabla u = 0$ for $p \leq 4$, regardless of the spatial dimension. Thus a solution strategy as described in [33] may be considered.

The shape optimization procedure is summarized in Algorithm 1, where we follow a standard approach via the Lagrange multiplier rule.

---

**Algorithm 1** Shape Optimization Procedure

---

1: $\Omega_0 \subset \mathbb{R}^d$
2: $k \leftarrow 0$
3: **repeat**
4:      Compute state $y^k$
5:      Compute adjoint variables $\hat{y}^k$
6:      Compute descent direction $u^k$ such that $j'(\Omega)u^k < 0$
7:      Choose $t_k > 0$ such that $j((\text{id} + t_k u^k)(\Omega)) < j(\Omega)$
8:      Set $\Omega_{k+1} = (\text{id} + t_k u^k)(\Omega)$
9:      $k \leftarrow k + 1$
10: **until** $j(\Omega_{k+1}) \leq j(\Omega_k)\, tol$

---

In a first step, the state $y$ is computed by solving the underlying boundary value problem which, in the present study, is given by the steady-state Navier–Stokes (16) below. In a second step the adjoint state $\hat{y}$ is computed which is associated with the Lagrange multipliers and given by the solution of the adjoint equations in (19). For the shape derivative $j'(\Omega)u$ we consider the boundary formulation (9) from which the descent direction $u$ is obtained by applying one of the three different strategies. The process to determine the modified shape deformation with $p$-Laplace extension is summarized in Algorithm 2.

A full non-linear approach is to solve the $p$-Laplace problem, that is, to solve (12). Algorithm 3 schematically illustrates the realization of this approach.

A rigorous comparison of the descent direction influence on the shape optimization procedure would require the determination of an optimal step size for each direction. However, the identification of the optimal step size is computationally demanding when the state is defined by the solution of a PDE, and thus this might be unfeasible. Nevertheless, we control the sequence of successive shape updates $k$ by applying

$$t_k = \frac{\alpha_k}{\max\limits_{x \in \Omega}\|u^k(x)\|_2}. \tag{13}$$

where $\alpha_k$ is chosen such that the *Armijo condition* is fulfilled.

---

**Algorithm 2** Steklov–Poincaré with $p$-Laplace domain extension (SP+p)

---

**Require:** $p_{max}, p_{inc}, \epsilon_1, \epsilon_2$
  1: Compute the boundary deformation $u$ according to (5)
  2: $\bar{u}_0 \leftarrow u$                             ▷ Use the solution from 5 as initial guess.
  3: $p \leftarrow 2$
  4: **repeat**
  5:     **if** $p < p_{max}$ **then**
  6:         $\epsilon \leftarrow \epsilon_1$
  7:     **else**
  8:         $\epsilon \leftarrow \epsilon_2$                     ▷ Where $\epsilon_2 << \epsilon_1$
  9:     **end if**
  10:    Compute the dom. extension according to (6) with initial guess $\bar{u}_0$ and tolerance $\epsilon$.
  11:    $p \leftarrow p + p_{inc}$
  12:    $\bar{u}_0 \leftarrow \bar{u}$                        ▷ Set the initial guess for the next $p$.
  13: **until** $p > p_{max}$
  14: $u \leftarrow \bar{u}$                          ▷ Set the deformation for the whole domain.

---

**Algorithm 3** $p$-Laplace relaxed steepest descent direction

---

**Require:** $p_{max}, p_{inc}, \epsilon_1, \epsilon_2$
  1: $u_0 \leftarrow 0$
  2: $p \leftarrow 2$
  3: **repeat**
  4:     **if** $p < p_{max}$ **then**
  5:         $\epsilon \leftarrow \epsilon_1$
  6:     **else**
  7:         $\epsilon \leftarrow \epsilon_2$                     ▷ Where $\epsilon_2 << \epsilon_1$
  8:     **end if**
  9:    Compute the deformation field according to (12), initial guess $u_0$ and tolerance $\epsilon$.
  10:    $p \leftarrow p + p_{inc}$
  11:    $u_0 \leftarrow u$                         ▷ Set the initial guess for the next $p$.
  12: **until** $p > p_{max}$

---

Furthermore, an additional issue that one might face in CAD-free shape optimization relates to the construction of the problem. In general but most frequently in internal flow shape optimization problems, such as the ones studied in this paper, we are interested in optimizing a certain Section of the wall, namely $\Gamma_D \subset \Gamma_W$. We thus define $\Gamma_W := \Gamma_D \cup \Gamma_B$ as the union of the non-intersecting sets of design and non-design (bound to their initial configuration) points, respectively. The construction of the optimization problem results in a change of the boundary condition of $u$ in $\Gamma_W$. A common problem that might arise, is that the sudden change of boundary conditions and displacement leads to distorted computational grids, as shown in Figure 1 (right). To ensure compatibility, we apply a filtering approach in a close neighborhood around the connection of $\Gamma_D$ and $\Gamma_B$, that reads

$$u_f(x) = \begin{cases} u(x) \frac{1}{2}\left(1 - \cos\left(\pi \frac{r(x)}{r_0}\right)\right), & \text{if } r(x) \leq r_0 \\ u(x), & \text{otherwise,} \end{cases} \tag{14}$$

where $r_0$ controls the filtering radius with

$$r(x) = \sqrt{(x_1 - \bar{x}_1)^2 + (x_2 - \bar{x}_2)^2}. \tag{15}$$

For the application studied herein, $\bar{x} = (\bar{x}_1, \bar{x}_2)$ corresponds to the position vector of a node connecting $\Gamma_D$ and $\Gamma_B$. Figure 1 schematically shows the impact of the filter for the same 2D case. As shown in Figure 1 (left), the optimizer has managed to update the shape while maintaining its grid quality. In contrast, when the solution $u$ is directly applied, the

grid quality is rapidly deteriorated, leading to even intersecting faces as shown in Figure 1 (right), making the numerical solution of the PDE constraints unfeasible.

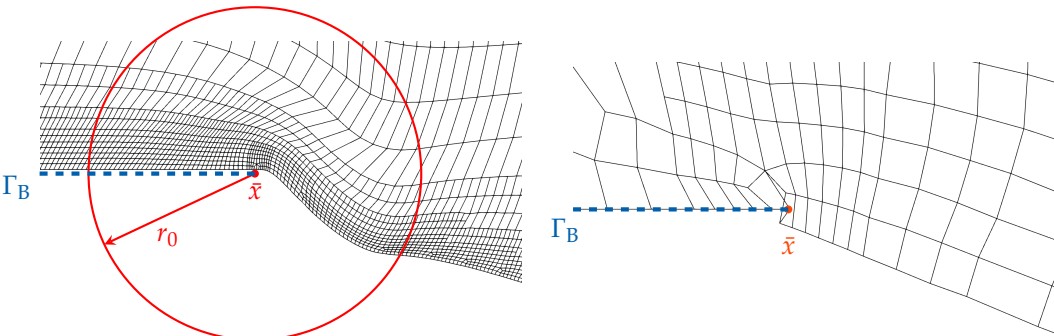

**Figure 1.** Two-dimensional optimization case: Detail of the computational unstructured grid of intermediate updated shapes around the connection of $\Gamma_D$ and $\Gamma_B$ using (**left**) $u_f$ and (**right**) $u$.

### 2.3. Governing Equations

Here, we consider the domain $\Omega \subset \mathbb{R}^2$, and the state is described by the velocity $v_i : \Omega \to \mathbb{R}$, $i = 1, 2$ and pressure $p : \Omega \to \mathbb{R}$ which are the solution to the stationary Navier–Stokes equations of an incompressible, Newtonian fluid, viz.

$$
\begin{aligned}
-\frac{\partial v_i}{\partial x_i} &= 0 && \text{in } \Omega, \\
-\frac{\partial}{\partial x_j}\left(\nu\left(\frac{\partial v_i}{\partial x_j} + \frac{\partial v_j}{\partial x_i}\right)\right) + v_j\frac{\partial v_i}{\partial x_j} &= -\frac{1}{\rho}\frac{\partial p}{\partial x_i} && \text{in } \Omega, \\
v_i &= 0 && \text{on } \Gamma_W, \\
v_i &= v_{i,\text{in}} && \text{on } \Gamma_I, \\
\nu\left(\frac{\partial v_i}{\partial x_j} + \frac{\partial v_j}{\partial x_i}\right)n_j &= pn_i && \text{on } \Gamma_O, \\
i &= 1, 2 \,,
\end{aligned}
\tag{16}
$$

where $\nu > 0$ is the kinematic viscosity and $\rho > 0$ denotes the density of the fluid. The aim of the present shape optimization procedure is to minimize the power loss within the flow domain which is described by the objective function

$$
J(\Gamma, v_i, p) = -\int_\Gamma \left(p + \frac{\rho}{2}v_i^2\right)v_j n_j \, ds \,.
\tag{17}
$$

By introducing the multipliers $\hat{v}_i$, $i = 1, 2$ and $\hat{p}$ we define the Lagrangian

$$
\begin{aligned}
L(\Gamma, (v, p), (\hat{v}, \hat{p})) &:= \int_\Gamma \left(p + \frac{\rho}{2}v_i^2\right)v_j n_j \, ds \\
&+ \int_\Omega -\nu\left(\frac{\partial v_i}{\partial x_j} + \frac{\partial v_j}{\partial x_i}\right)\frac{\partial \hat{v}_i}{\partial x_j} + v_j\frac{\partial v_i}{\partial x_j}\hat{v}_i - \frac{1}{\rho}p\frac{\partial \hat{v}_j}{\partial x_j} - \frac{\partial v_j}{\partial x_j}\hat{p} \, dx,
\end{aligned}
\tag{18}
$$

which for arbitrary $\hat{y} = (\hat{v}, \hat{p})$ gives $j(\Gamma) = J(\Gamma, (v, p)(\Gamma)) = L(\Gamma, (v, p)(\Gamma), (\hat{v}, \hat{p}))$ if the state Equation (16) is fulfilled. The Lagrange multipliers are identified with the adjoint state $\hat{y} = (\hat{v}, \hat{p})$ which is the solution to the system of adjoint equations to (16). The system of adjoint equations defined on the reference domain read

$$-\frac{\partial \hat{v}_i}{\partial x_i} = 0 \qquad \text{in } \Omega,$$

$$-\frac{\partial}{\partial x_j}\left(\nu\left(\frac{\partial \hat{v}_i}{\partial x_j} + \frac{\partial \hat{v}_j}{\partial x_i}\right)\right) - \frac{\partial \hat{v}_i}{\partial x_j}v_j + \frac{\partial v_j}{\partial x_i}\hat{v}_j + \frac{1}{\rho}\frac{\partial \hat{p}}{\partial x_i} = 0 \qquad \text{in } \Omega,$$

$$\hat{v}_i = 0 \qquad \text{on } \Gamma_{\text{W}}, \quad (19)$$

$$\hat{v}_i\, n_i = v_i\, n_i \qquad \text{on } \Gamma_{\text{I}},$$

$$\hat{p} = \hat{v}_n\, v_n - \frac{1}{2}\rho v_i^2 - \rho v_n^2 \qquad \text{on } \Gamma_{\text{O}},$$

$$i = 1, 2\,.$$

With the solution of the primal and adjoint equations, one can compute the shape sensitivity given by the expression

$$j'(\Gamma)u = \int_{\Gamma_{\text{D}}} \underbrace{-\mu\left(\frac{\partial v_i}{\partial n}\frac{\partial \hat{v}_i}{\partial n}\right)}_{=\sigma} u_j n_j\, ds. \qquad (20)$$

*2.4. Numerical Method*

The numerical procedure for the solution of the primal (16) and adjoint system (19) is based upon the finite volume method (FVM) [9,34]. The implicit numerical approximation is second-order accurate in space and time and supports arbitrary polyhedral cells as well as local grid refinement. The segregated algorithm uses a cell-centered, co-located storage arrangement for all transport properties. A detailed derivation of this hybrid adjoint approach can be found in [6,9,11]. The primal and adjoint pressure–velocity coupling utilizes a pressure-correction scheme and parallelization is realized by means of a domain decomposition approach [35,36].

**3. Application**

The application considered refers to a 2D S-bent duct. Results compare three different concurrent mesh and shape update approaches, i.e., a Steklov–Poincaré (SP) method, a Steklov–Poincaré with a subsequent $p$-Laplace extension (SP+$p$) and a $p$-Laplace approach. In both methods we choose the value of $p_{max} = 4.1$. This value is slightly above the threshold value $p = 4$ that is deemed to by large enough as described in Section 2. The SP approach essentially follows from (5) with the inner product (7) and the SP+$p$ method is described by Algorithm 2. The $p$-Laplace mesh and shape update approach corresponds to the original $p$-Laplace method outlined in Algorithm 3. Emphasis is put on assessing the evolution of the objective functional, the final shape, the computational cost, and the preservation of the grid orthogonality and the aspect ratio.

*3.1. Two-Dimensional Bent*

The application considers the optimization of a two-dimensional S-bent duct. A sketch of the initial shape and the employed unstructured grid are shown in Figure 2. The total length of the computational domain is $L$ while $h_1 + h_2$ is its height. We assume wall sections of length $l$ from the inlet and outlet, where the shape is bound to its initial configuration. The following non-dimensional geometric ratios hold: $l/h_1 = 2$, $h_2/h_1 = 1.5$, $L/h_1 = 7.5$.

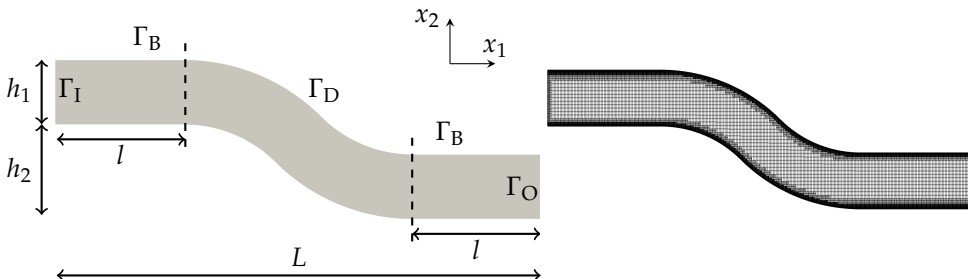

**Figure 2.** Illustration of the S-bent 2D duct's initial shape with geometric/boundary annotations (**left**) and employed unstructured, locally refined grid (**right**). The lower wall of the duct is split in $\Gamma_B$ and $\Gamma_D$ in accordance with the split of the upper wall.

To ensure the independence of the objective function $J$ with respect to the spatial discretization, a mesh sensitivity study is initially conducted. The results of the study are presented in Table 1. Since the estimated objective function doesn't change by more than 2% from refinement level 2 to 3, we employ an unstructured-grid discretization of 31,600 quadrilateral control volumes. As indicated by Figure 2 (right), the grid is progressively refined towards the wall boundaries. It features almost orthogonal control volumes with an approximate unity cell aspect ratio in the boundary-adjacent cell layer. The filtering approach described in Section 2 is applied around the four points, connecting $\Gamma_B$ and $\Gamma_D$ boundaries, with a normalized filtering radius of $r_0/h_1 = 0.2$. We assume a unidirectional, parabolic inlet velocity profile, that reads

$$v_{1,\text{in}}(x_2) = 2\, v_{ref}\left(1 - \left(\frac{2x_2}{h_1}\right)^2\right), \quad v_{2,\text{in}}(x_2) = 0, \tag{21}$$

where the coordinate origin aligns with the midpoint of $\Gamma_I$. The laminar flow is characterized by a Reynolds number of $\text{Re} = (v_{ref}\, h_1)/\nu = 500$.

**Table 1.** Mesh sensitivity study. Control volumes are abbreviated by CVs.

| Refinement Level | Number of CVs | $\frac{2J}{\rho v_{ref}^3 h_1}$ |
|:---:|:---:|:---:|
| 0 | 3250 | 1.06 |
| 1 | 16,800 | 0.97 |
| 2 | 31,600 | 0.89 |
| 3 | 44,130 | 0.88 |

3.1.1. Shape and Objective Functional Evolution

Figure 3 (left) displays the computed reduction of the normalized power-loss objective functional by about 18% from its initial value. Figure 3 (right) depicts the corresponding evolution of the shape different from the initial shape. It is observed that all approaches converge remarkably fast toward the final shape. A decrease of the regularization parameter $\eta_{max}$ in (8), to unity reveals a substantial change of deformation and thus the final shape, cf. Figure 3 (right).

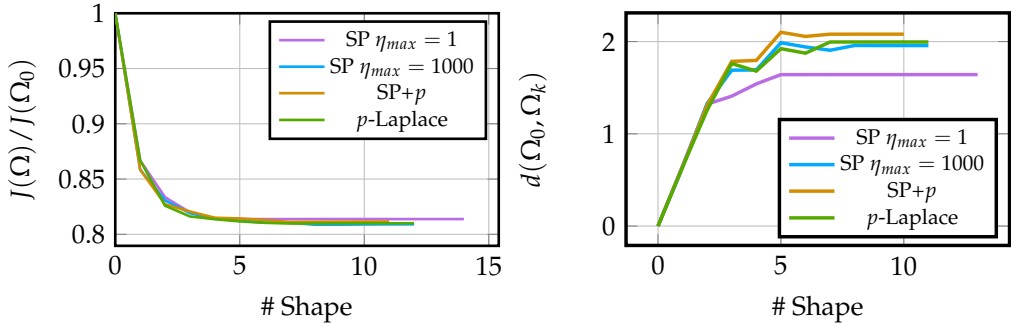

**Figure 3.** Evolution of the normalized objective functional for the 2D S-bent (**left**) and evolution of the symmetric shape difference $d(\Omega_0, \Omega_k) := \int_{\Omega_0 \setminus \Omega_k} 1 \, dx + \int_{\Omega_k \setminus \Omega_0} 1 \, dx$ (**right**).

For each final shape, obtained by the different optimization approaches, the objective is validated by a grid refinement study. Therefore the objective, computed on the deformed grid, is compared to the objective computed on newly generated grids at three different levels of refinement. Table 2 summarizes the results of the grid study where the third column shows the objective functional $J(\Omega_n)$ evaluated on the final shape $\Omega_n$ after $n$ optimization steps (cf. Table 3), relative to objective $J_i(\Omega_n)$ evaluated on newly generated grids for the refinement levels $i = 1, 2, 3$. The results show that the difference is at most 2% or 3%, respectively, for the individual cases.

**Table 2.** Mesh sensitivity study for the optimized shapes; where $J$ is the approximate objective value computed on the grid which is used for the optimization, and $J_i$ is computed after remeshing with different refinements.

| Refinement Level $i$ | Number of CVs | $|J/J_i|$ |
|:---:|:---:|:---:|
| \multicolumn{3}{c}{SP $\eta_{max} = 1$} |
| 1 | 16,297 | 0.99 |
| 2 | 33,735 | 1.01 |
| 3 | 53,080 | 1.02 |
| \multicolumn{3}{c}{SP $\eta_{max} = 1000$} |
| 1 | 16,130 | 0.99 |
| 2 | 33,859 | 1.01 |
| 3 | 52,405 | 1.02 |
| \multicolumn{3}{c}{SP+$p$} |
| 1 | 16,172 | 0.99 |
| 2 | 33,658 | 1.01 |
| 3 | 51,723 | 1.03 |
| \multicolumn{3}{c}{$p$-Laplace} |
| 1 | 16,172 | 0.99 |
| 2 | 33,658 | 1.01 |
| 3 | 51,723 | 1.03 |

Figure 4 displays the final shapes predicted by the four optimizations. Minor differences between the respective shapes returned by the $p$-Laplace, the SP+$p$, and SP ($\eta_{max} = 1000$) methods, are observed. Except for the transition region, where the deformed part of the boundary meets the fixed part, the shapes obtained by all approaches reduce the curvature of the boundary and straighten the bent part. At the lower left and upper right transition region, the optimized shapes tend to form a kink in the shape of a step. Apart from the SP approach with $\eta_m max = 1$ the final shapes are rather close and the contours of the shape almost cover each other.

**Table 3.** Total number of optimization steps for the 2D S-bent, normalized average wall-clock time per design step required to compute the descent direction, and normalized convergence effort. All normalization computed with results of the SP approach (p-Laplace approaches employed $p = 4.1$).

| 2D S-Bend | Steps | Norm. Time/Step | Norm. Effort |
|---|---|---|---|
| Steklov-Poincaré (SP; $\eta_{max} = 1000$) | 11 | 1 | 1 |
| Steklov-Poincaré (SP; $\eta_{max} = 1$) | 5 | 2.4 | 1.1 |
| Steklov-Poincaré with $p$-Laplace ext. (SP+p) | 10 | 2.8 | 2.6 |
| $p$-Laplace | 9 | 4.3 | 3.6 |

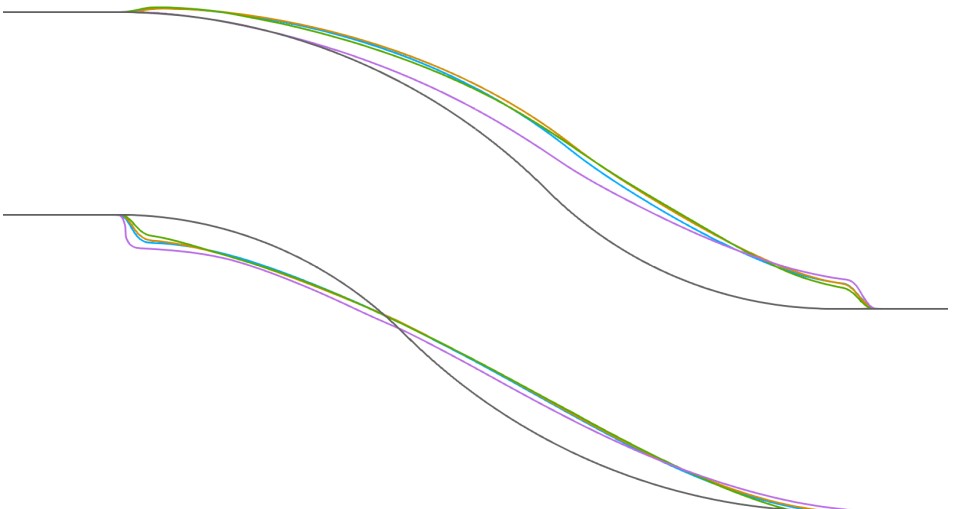

**Figure 4.** Contours of the final shapes for the 2D S-bent optimization displayed by the Initial shape ———, Steklov–Poincaré ——— with $\eta_{max} = 1$ and ——— with $\eta_{max} = 1000$, the Steklov–Poincaré with $p$-Laplace extension ——— and the $p$-Laplace ——— methods.

Figure 5 displays the distribution of the objective function along the inlet and outlet for the initial and optimized shape obtained from the $p$-Laplace method (left) and the respective drop of the absolute values from initial to optimized (right). The computed values follow from $j_\Gamma(\Gamma_I) = j_\Gamma(0, x_2)$ and $j_\Gamma(\Gamma_O) = j_\Gamma(L, x_2 - h_2)$ where $j_\Gamma(\Gamma)$ denotes the integrand of Equation (17). We remind the reader that $(x_1, x_2) = (0, 0)$ lies at the midpoint of $\Gamma_I$. Along the inlet, where the velocity values are fixed, a parabolic drop of the absolute objective value is observed, which is due to the homogeneous decrease of the pressure value, as illustrated in Figure 6. Along the outlet, the pressure is fixed and the velocities of the optimized shape are more homogeneous as compared to the initial shape. Smaller core flow velocities are confirmed for the optimal shape in Figures 7 and 8 and yield a reduction of the power loss, cf. Figure 5. In theory, an unbounded increase of the duct's area (volume) would result in a minimization of the power loss within the flow domain. However, by restricting certain sections of the shape to their initial configuration, an optimal solution is not simply associated with an area increase, as also displayed by Figure 4. Due to the constrained segments near the inlet and outlet of the domain, an area increase would promote sudden expansion/contraction losses due to recirculation and drive the total pressure loss away from an optimal solution. On the other hand, the flow on the optimized shapes herein, admits no recirculation regions while at the same time minimizing the velocity magnitude at the core of the flow as illustrated in Figure 8.

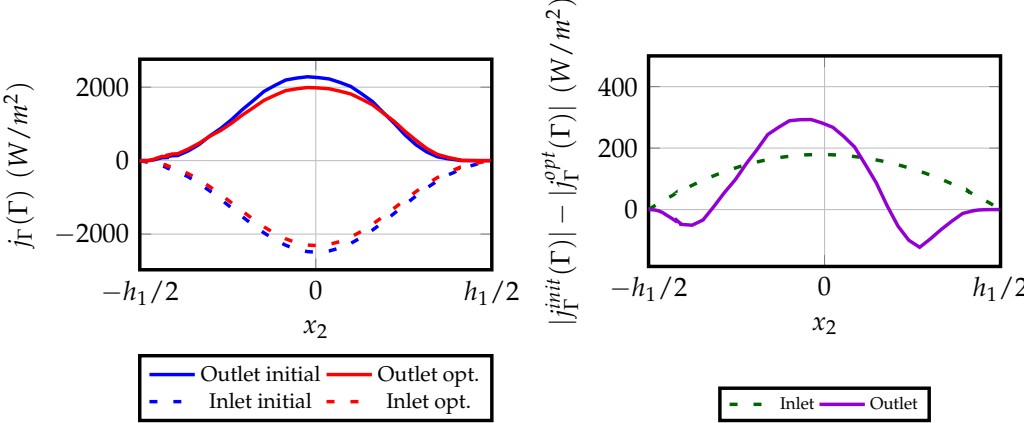

**Figure 5. Left**: The distribution of the objective function (17) along the inlet (dashed lines) and outlet (continuous lines) boundaries for the initial (blue) and the optimized (p-Laplace, red) shape of the 2D S-bent. Here, $-h_1/2$ refers to the bottom corner of each boundary. **Right**: Distribution of the drop of the absolute objective function from initial ($j_\Gamma^{init}$) to optimized ($j_\Gamma^{opt}$) shape along the inlet and outlet.

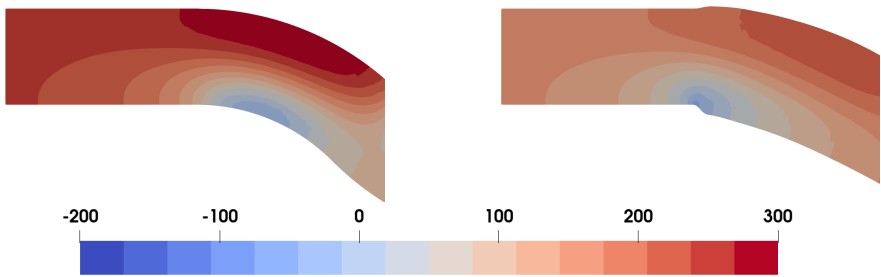

**Figure 6.** Pressure contours in [Pa] near the inlet for the initial (**left**) and optimized (**right**; p-Laplace) shapes of the 2D S.bent. The exit pressure is assigned to zero in both cases.

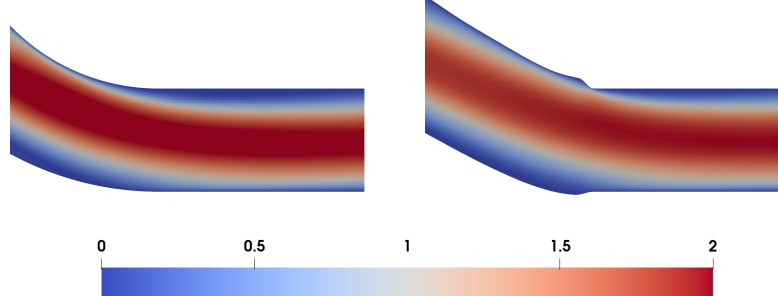

**Figure 7.** Contours of the velocity magnitude in [m/s] near the outlet for the initial (**left**) and optimized (p-Laplace) (**right**) shapes of the 2D S-bent.

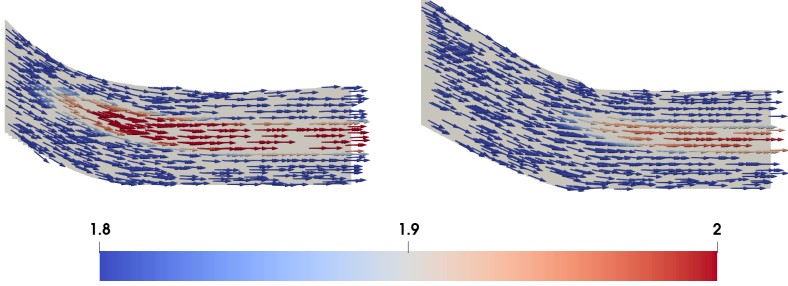

**Figure 8.** Velocity magnitude vectors in [m/s] near the outlet for the initial (**left**) and optimized (p-Laplace) (**right**) shapes of the 2D S-bent.

Table 3 provides information on the computational efforts. It is seen that the severe non-linearity inherent to both *p*-Laplace approaches significantly increases the computational cost. Compared to the SP approaches, the SP+*p* method increases the effort approximately by a factor of 2.5, whereas the *p*-Laplace method is afflicted with roughly 3.5 times the costs of the SP approach. Since the optimization convergence varies, these average costs per design step were normalized by the convergence ratio. We also observed that the lower the $\eta_{max}$ value is, the more computationally expensive the SP approach becomes. These costs will usually be compared to the flow simulation efforts, which are rapidly increasing for more complex flows and can substantially benefit from an improved mesh quality. Moreover, the credibility of the optimization result is an issue that is usually assessed by comparing the objective functional for the optimized geometry obtained in conjunction with the morphed and a new mesh. In this regard, an improved mesh quality might speed up the convergence and the feasibility of a morphing-based optimization.

### 3.1.2. Mesh Quality

With attention directed to the mesh quality, Figure 4 also indicates that all methods display the most severe deformation where the design wall meets the non-design wall, which might also challenge the attainable mesh quality in this region. Figures 9, 10 and Table 4 reveal that the SP+*p* method can lift the cell minimum angle in comparison to the SP method ($\eta_{max} = 1000$) while also reducing the amount of inferior cells by one order of magnitude, and thereby improve the mesh quality. However, the *p*-Laplace method displays even more localized changes and smaller deviations from the ideal grid arrangement, cf. right graph of Figure 9. The critical lowest value refers to approximately 35° and the total amount of critical cells is reduced by approximately one [two] order[s] of magnitude in comparison to the SP+*p*[SP] method, cf. Figure 10. Interestingly, a reduction in the regularization parameter $\eta_{max} = 1000$ to an exemplary alternative value of $\eta_{max} = 1$ results in a considerable deterioration of the cell shapes, as indicated by the critical cells denoted in Table 4 and the histogram in Figure 10. The latter is also seen by an observation of the minimum orthogonality depicted in Figure 11, where a continuous deterioration of cell quality is observed with increasing deformation when the regularization parameter is reduced. To this extent, the benefits of the *p*-Laplace and the hybrid SP+*p* approach are outlined.

**Table 4.** Most critical grid-orthogonality values observed for morphed mesh of the final design of the 2D S-bent. Values indicate minimum of $90° − \beta$, where $\beta$ refers to the angle between a face normal and the connecting line between the adjacent cell centers.

| | Initial | SP ($\eta_{max} = 1$) | SP ($\eta_{max} = 1000$) | SP+p ($\eta_{max} = 1000, p = 4.1$) | p-Laplace ($p = 4.1$) |
|---|---|---|---|---|---|
| min. angle | 50 | 17 | 27 | 30 | 35 |

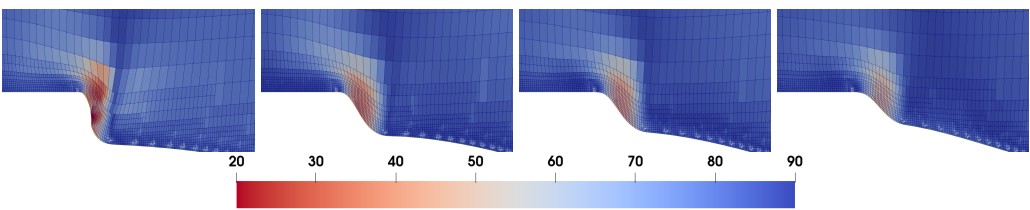

**Figure 9.** Grid orthogonality of the final design in the lower downstream transition region of the 2D S-bent. The displayed property is $90° − \beta$, where $\beta$ refers to the angle between a face normal and the connecting line between the adjacent cell centers and higher values (blue) are better. The results are obtained with, from left to right, SP with $\eta_{max} = 1$ and with $\eta_{max} = 1000$, SP+*p*, and *p*-Laplace approach.

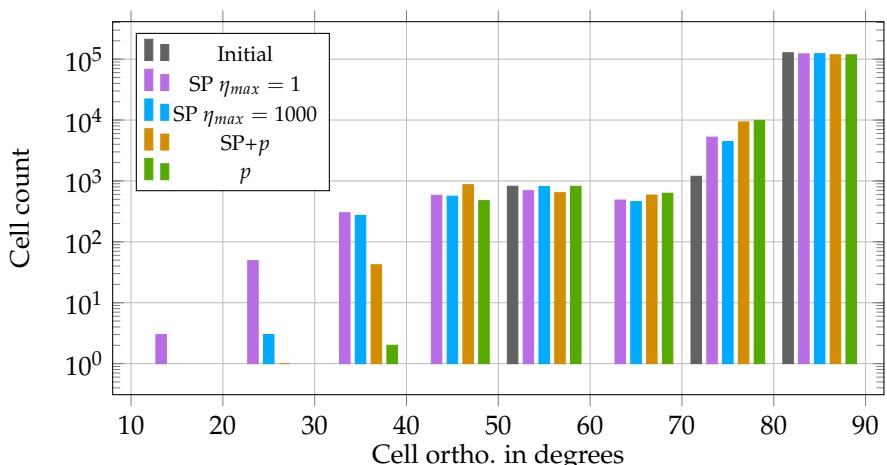

**Figure 10.** Grid-orthogonality observed for morphed mesh of the final design of the 2D S-bent. Distribution of $90° - \beta$, where $\beta$ refers to the angle between a face normal and the connecting line between the adjacent cell centers.

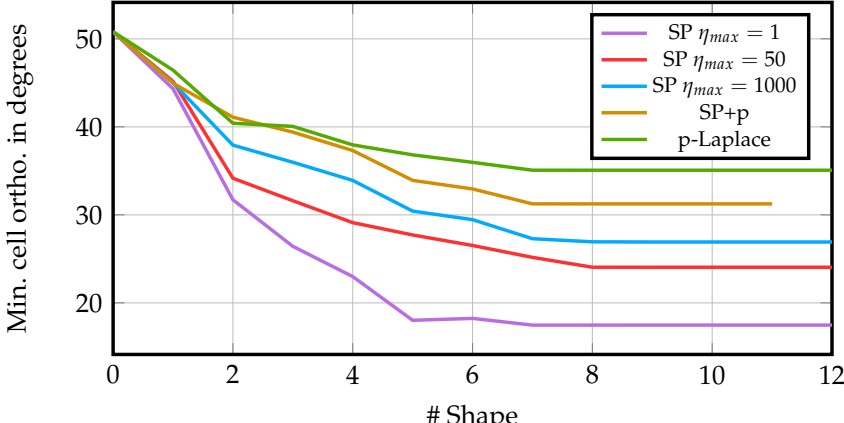

**Figure 11.** Evolution of the minimum orthogonality for the initial and deformed meshes during the optimization.

Figure 12 displays the cell aspect ratio in the vicinity of the transition regime for the final shapes of all three methods. Again, the leftmost graphs refer to the SP approach, followed by the SP+$p$ method and the p-Laplace method (rightmost graph). The deterioration of the aspect ratio is moderate and most pronounced by the SP method. Again, a $p$-Laplace extension helps to preserve the mesh quality, as depicted by the SP+$p$ results. Aspect ratios obtained from the $p$-Laplace method are in fair qualitative agreement with the SP+$p$ grids, though reported $p$-Laplace values are again superior. Furthermore, it is noted that an increasing value of $\eta_{max}$ enables the SP method to increase the overall quality of the produced mesh, without, however, being able to reach the beneficial behavior of the SP+$p$ and $p$-Laplace methods.

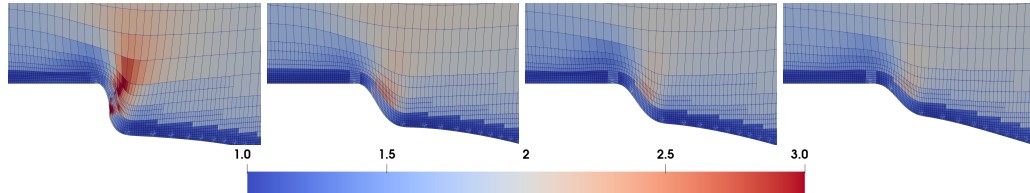

**Figure 12.** Cell aspect ratio at the lower channel for the final shapes each obtained from left to right with the SP with $\eta_{max} = 1$ and with $\eta_{max} = 1000$, the SP+$p$ and with the $p$-Laplace method for the optimized shapes of the 2D bent in the lower downstream transition region (range from 1 to 3 where lower values (blue) are better).

## 4. Conclusions

This paper investigated approaches to simultaneously compute the shape and mesh update in the context of a parameter-free, adjoint-assisted shape optimization. Included methods refer to Steklov–Poincaré (SP) and $p$-Laplace approaches and a newly proposed hybrid approach. The shape update of the hybrid approach follows from the SP method, while the extension to the domain is realized based on the $p$-Laplace method. The motivation of the newly proposed method (SP+$p$) was to extract the positive properties of both ingredients, with the lower computational cost of SP and the improved mesh quality preservation of $p$-Laplace. The implementation of the SP+$p$ approach was described in accordance with the established individual baseline methods, and technical aspects, such as the treatment of deformations near the connection of design and non-design boundaries, were discussed.

The suggested methods were applied to the power loss optimization in a 2D steady-state, laminar incompressible fluid flow. All approaches managed to locate an optimal solution after 5–10 shape updates. However, as regards the computational effort and mesh quality metrics, the observations verified our initial hypothesis. The SP+$p$ method performed better than a pure SP approach in terms of the quality of the optimized mesh while also requiring approximately half the computational effort of the $p$-Laplace approach. Additionally, we have studied the influence of the regularization parameter value for the diffusion coefficient of the SP approach on mesh quality metrics. Results indicate that the quality of the initial mesh is better preserved as the parameter increases. Nevertheless, for all investigated values, the mesh quality metrics of the SP approach fell short in comparison to the SP+$p$ and $p$-Laplace approach.

Future work will scrutinize the proposed hybrid method (SP+$p$) in three-dimensional geometries and external flows to investigate if the benefits of the method hold in these scenarios as well.

**Author Contributions:** Conceptualization, P.M.M., G.B., and T.R.; methodology, P.M.M., G.B., and T.R.; validation, P.M.M. and G.B.; formal analysis, P.M.M. and G.B.; investigation, P.M.M. and G.B.; resources, T.R.; writing—original draft preparation, P.M.M., G.B., and T.R.; writing—review and editing, P.M.M., G.B., and T.R.; visualization, P.M.M. and G.B.; supervision, T.R.; project administration, T.R.; funding acquisition, T.R. All authors have read and agreed to the published version of the manuscript.

**Funding:** The current work is a part of the research training group 'Simulation-Based Design Optimization of Dynamic Systems Under Uncertainties' (SENSUS) funded by the state of Hamburg within the Landesforschungsförderung under project number LFF-GK11, and the Research Training Group RTG 2583 'Modeling, Simulation and Optimization of Fluid Dynamic Applications' support by the Deutsche Forschungsgemeinschaft (DFG). Publishing fees supported by Funding Programme Open Access Publishing of Hamburg University of Technology (TUHH).

**Conflicts of Interest:** The authors declare no conflict of interest.

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
