# Peer review of "Shape Transformation Approaches for Fluid Dynamic Optimization"

_aerospace, doi:10.3390/aerospace10060519_

Round 1

Reviewer 1 Report

The authors investigated various approaches to simultaneously compute the shape and mesh update in the context of a parameter-free, adjoint-assisted shape optimization. The presented results are interesting, and the paper has an excellent scientific soundness.

The main quantitative findings are to be mentioned in the abstract.

The introduction is relatively short and may be extended.

The Mathematical and Computational Model are very well established and presented.

can the numerical method be used for 3D geometries? What will be the computational cost in this case?

Can the present numerical method be used for time dependent problems?

For the presented applications, it will be interesting to perform a mesh sensitivity test.

In addition to velocity contours, it will be interesting to present the streamlines, for a better understanding of the flow structure.

Reviewer 2 Report

The paper is about combined shape- and mesh-update strategies for parameter free shape optimization methods. Three different strategies to translate the shape sensitivities computed by adjoint shape optimization procedures into simultaneous updates of both the shape and the discretized domain are employed in combination with a mesh-morphing strategy. The methods that are considered by the authors  involve a linear Steklov-Poincaré (Hilbert-space) approach, a  highly non-linear p-Laplace (Banach-space) method as well as a hybrid variant which updates the shape in Hilbert space. These methods are scrutinized for optimizing the power loss of a two-dimensional bended duct flow using an unstructured, locally refined grid that initially displays favorable grid properties. The authors compare the optimization results with respect to the optimization convergence, the computational effort and, moreover, the preservation of the mesh quality during the optimization sequence.
The subject of the article is important both from the mathematical and applied points of view. I believe that the obtained results give an interesting contribution in the subject area. Other than that, the paper is well written, the problems and methods involved are carefully presented and analyzed.
The manuscript can be considered for publication provided that the authors revise the paper based on the above suggestions.
1) The authors should refine the main innovation of the article in the introduction. What essentially makes your study different from existing literature?
2) The physical meaning of the results can also be more discussed. This will help readers better understand the results as well as their implication.
3) The paper includes relevant literature in the field. However, the authors may enriche the introduction section and additionally emphasize the importance of topic by consideration the following papers on fluid dynamic optimization in Hilbert and/or Banach spaces formulation:
Fursikov, A.V.  Flow of a viscous incompressible fluid around a body: boundary-value problems and minimization of the work of a fluid. J Math Sci. 2012, 180, 763-816, https://doi.org/10.1007/s10958-012-0670-1
Baranovskii, E.S. Optimal boundary control of nonlinear-viscous fluid flows. Sb. Math. 2020, 211, 505-520, https://doi.org/10.1070/SM9246
4) Future recommendations should be added to help other investigators expand on the presented exploratory analysis.

Reviewer 3 Report

This paper, aerospace-2401713, deals with shape optimization using various mathematical approaches.

In my opinion, this is a very high quality item. The English language is very good. The literature review is well done. I recommend this paper for publication as is. I only have one second comment:

I think something is missing in figure 4. Can you please correct this figure?

Round 2

Reviewer 2 Report

I am satisfied with the answer of the authors to my comments from the first review. The points that were initially unclear are now clear. 
I congratulate the authors for obtaining excellent results.
This article can be recommended for publication in the current form.